# Diagnostic Methods for Evaluation of Gastric Motility—A Mini Review

**DOI:** 10.3390/diagnostics13040803

**Published:** 2023-02-20

**Authors:** Yan Wang, Jiande D. Z. Chen, Borko Nojkov

**Affiliations:** Division of Gastroenterology, University of Michigan, Ann Arbor, MI 48109, USA

**Keywords:** functional dyspepsia, gastroparesis, gastric motility, antroduodenal contraction, gastric emptying, gastric slow waves, electrogastrography

## Abstract

Gastric motility abnormalities are common in patients with disorders of gut-brain interaction, such as functional dyspepsia and gastroparesis. Accurate assessment of the gastric motility in these common disorders can help understand the underlying pathophysiology and guide effective treatment. A variety of clinically applicable diagnostic methods have been developed to objectively evaluate the presence of gastric dysmotility, including tests of gastric accommodation, antroduodenal motility, gastric emptying, and gastric myoelectrical activity. The aim of this mini review is to summarize the advances in clinically available diagnostic methods for evaluation of gastric motility and describe the advantages and disadvantages of each test.

## 1. Introduction

Among disorders of gut-brain interaction (DGBI), functional dyspepsia (FD) is one of the most common entities, with an estimated prevalence of approximately 16% in the general population [1]. FD is diagnosed based on typical symptoms (epigastric pain or burning, postprandial fullness, and/or early satiation) and the absence of underlying structural disease to explain these symptoms [2]. FD is a chronic condition, and, per the Rome expert consensus criteria, symptoms must be present for at least 6 months to establish the diagnosis [2]. Based on the predominant symptoms, patients with FD are sub-classified into three diagnostic categories: (1) Postprandial distress syndrome (PDS), characterized by postprandial nature of symptoms, (2) Epigastric pain syndrome (EPS), characterized by abdominal pain as predominant symptom, and (3) patients with overlapping PDS and EPS symptoms (PDS/EPS).

Gastroparesis (GP) is a DGBI that is defined by impairment of gastric motility resulting in prolonged food retention in the stomach and the presence of associated symptoms [3]. Patients with GP typically have nausea, vomiting, early satiety, and postprandial fullness, although other symptoms such as epigastric pain, bloating and belching are also common. The diagnosis of GP requires objective verification of a delayed gastric emptying in absence of mechanical obstruction of the gastrointestinal tract [4]. The global population-based prevalence of gastroparesis is estimated at approximately 1.4% [5].

Both FD and gastroparesis are disorders associated with high clinical burden, including a negative impact on the quality of life and high associated financial costs [6].

Patients with FD and gastroparesis present with common symptoms, including postprandial fullness, early satiation, epigastric pain, bloating, nausea, and vomiting. Postprandial fullness, early satiation, and epigastric pain were recognized as cardinal FD symptoms, whereas nausea and vomiting were not, allowing the symptom profiles of FD and gastroparesis to be discriminated [7]. FD and gastroparesis also differ in how they are diagnosed. While the diagnosis of FD is established based on fulfillment of symptoms criteria in absence of structural disease, the diagnosis of gastroparesis requires diagnostic testing to confirm a delay in the gastric emptying [2]. These disorders were also found to have an overlapping pathophysiology. Pathophysiological mechanisms, relevant in both FD and gastroparesis, that are associated with gastric dysmotility include impaired gastric accommodation, antral hypomotility, delayed gastric emptying, and gastric electrical dysrhythmia [8].

Gastric accommodation is a crucial function that allows adequate relaxation of the fundus and the proximal gastric body. It is induced by food ingestion and regulated by the nitric oxide-mediated vagal reflex [9]. Gastric accommodation is also influenced by antro-fundic reflex relaxation in response to antral distension. Impairment of the gastric accommodation was found in 40% of patients with FD and 43% of patients with idiopathic gastroparesis [10,11]. An injury to the vagus nerve or autonomic dysfunction could lead to impaired gastric accommodation and explain the sensation of early satiety or postprandial fullness in patients with FD and gastroparesis [12].

Antral hypomotility is the most common form of gastric motor dysfunction when patients with FD and gastroparesis are evaluated using antroduodenal manometry [13]. However, it remains unknown whether antral hypomotility is a primary mechanism, or whether decreased frequency of antral contractions is secondary to the underlying antral distension and impaired gastric accommodation. Interstitial cells of Cajal (ICCs) are crucial for regulation of smooth muscle contractility as they initiate and propagate the gastric electrical slow waves [14]. Histopathological studies of full-thickness gastric biopsies in patients with severe diabetic and idiopathic gastroparesis found a reduction of ICCs, providing a cellular basis to explain the antral hypomotility [15].

Delayed gastric emptying is the hallmark finding required to diagnose gastroparesis. Approximately 35% of unselected patients with FD will also have delayed gastric emptying, if evaluated [1,2]. The pathophysiology of delayed gastric emptying is complex and includes alterations in gastric accommodation, gastric motor function, and abnormalities in pyloric function and duodenal motility [16]. The relationship between gastric emptying delay and dyspeptic symptoms remains poorly understood as studies consistently failed to show that the gastric emptying delay is an indicator of dyspeptic symptoms severity [17]. However, more recent data from patients with gastroparesis revealed that, when investigated with adequate diagnostic testing, delayed gastric emptying correlates with the severity of dyspeptic symptoms [18]. Furthermore, certain prokinetic medications, such as domperidone and relamorelin, were found effective to both improve gastric emptying and alleviate upper gastrointestinal symptoms [19].

Altered gastric myoelectrical activity was reported in two-thirds of patients with FD when evaluated with noninvasive electrogastrography (EGG) [20]. EGG alterations indicative of altered gastric electrical slow waves were also found in majority of evaluated patients with gastroparesis [21]. Furthermore, EGG alterations were linked with nausea, a hallmark symptom of gastroparesis [22]. Gastric electrical dysrhythmia, discovered using high-resolution (HR) EGG, was also found to be associated with nausea and vomiting syndromes in a recent study by Gharibans et al. [23]. In a study of gastroparesis patients treated with the prokinetic medication cisapride, patients who normalized their EGG alterations exhibited a higher gastric emptying rate compared to patients with persistently abnormal gastric electrical rhythm despite cisapride [24]. The abnormal gastric myoelectrical activity in patients with gastroparesis or chronic nausea and vomiting can be partly explained by depletion of ICCs that generate and propagate gastric electrical slow waves [25,26,27]. In aggregate, these findings suggest that altered gastric myoelectrical activity (“gastric dysrhythmias”) seem to have a crucial role in the pathophysiology of FD and gastroparesis.

## 2. Assessment of Gastric Accommodation

### 2.1. Barostat

The gastric barostat is a device that currently represents the gold standard for assessment of gastric accommodation. It consists of a polyethylene balloon, with a volume of up to 1.0–1.2 L, connected to the barostat device through a double-lumen polyvinyl tube (Figure 1) [28]. The principle of a barostat is that it provides a constant pressure in the balloon via a built-in computerized pneumatic pump. The finely folded adherent balloon is perorally introduced into the stomach via the esophagus, attached with adhesive tape on the participant’s chin, and positioned to record volume change at a constant pressure in the proximal stomach. By maintaining a constant pressure inside the balloon and observing changes in balloon volume that correlate to changes in gastric volume, gastric volumes can be measured. The balloon pressure can be modified in a progressive or random manner, allowing for evaluation of the gastric sensation.

Impaired gastric accommodation determined by barostat testing was found to be frequently present in patients with non-ulcer dyspepsia and, while not affected by *Helicobacter pylori* infection, impaired accommodation was linked to increased visceral sensitivity [29]. Barostat testing has also been used to assess the mechanism of action of pharmacological therapy in patients with FD. For example, Tack and colleagues found buspirone to significantly improve FD symptoms after 4 weeks of therapy by improving gastric accommodation, which was assessed with a barostat [30].

The advantage of barostat is that it is a well-validated instrument capable of directly measuring volume changes in the proximal stomach [31]. For example, utilizing gastric barostat, Tack and colleagues found impaired gastric accommodation in 40% of patients with FD, which was related to symptoms of early satiety and weight loss [10]. Similarly, in another study, barostat testing found impaired gastric accommodation in 43% of patients with idiopathic gastroparesis [11]. However, the barostat method has its limitations. It requires significant time to complete the testing in addition to dedicated equipment and expertise. Moreover, it is invasive and often intolerable for patients [31]. The gastric balloon has also been reported to alter the distribution of gastric content and affect normal gastric physiology without impairing the gastric emptying [32]. Furthermore, temporary ectopic propagation and gastric electrical dysrhythmias were recently found to be induced by gastric distension with a barostat device [33]. However, the barostat remains an excellent method to investigate gastric accommodation and sensation, including evaluation of how various therapeutic interventions affect the gastric physiology.

### 2.2. Satiation Drinking Test

Satiation drinking test is a noninvasive tool developed to induce gastric distension and indirectly measure the postprandial sensorimotor functions, including gastric accommodation [34]. In this test, individuals drink a liquid of known composition and/or caloric content at a certain rate over pre-determined time-period. When maximum satiety is reached, the volume of ingested liquid is recorded to estimate the gastric accommodation and sensation [35]. A variety of satiation drinking test protocols has been reported, using either water or specific nutrient solutions [35]. Drinking test protocols also differ in rapidity of the liquid intake and the tested individual’s awareness of the quantity of ingestion.

Studies utilizing the satiation drink test consistently show that patients with FD and gastroparesis are able to drink significantly less volume compared to healthy controls and they report more severe associated symptoms [34,36]. The satiation drink test was recently proposed as a diagnostic biomarker for patients with FD, given its non-invasiveness, good tolerance, and reproducibility [37]. Furthermore, a study using a liquid meal slow drinking protocol found correlation between calorie intake and gastric accommodation in FD patients. In this study, the satiation drinking test was predictive of impairment in gastric accommodation with sensitivity and specificity of 92% and 86%, respectively [38].

The satiety drinking test was also used to evaluate the therapeutic response to certain pharmacological treatments in patients with FD [39,40]. This limited data indicates that satiation drinking tests may be predictive of the outcome of evaluated therapies (e.g., acotiamide, nortriptyline). However, more studies are needed to further evaluate the ability of satiety drinking test to serve as a predictive biomarker of treatment outcome.

The limitation of the satiety drinking test is that it is indirect, non-standardized, and controversial [31]. Gonenne J et al. found that in 85 healthy controls and 35 patients with FD, the drinking test could only explain 13% and 3% of the variation in fasting and postprandial volumes as measured by SPECT [41]. Other factors that affect gastric volume include nerve innervation, hormones, and gastric viscoelastic characteristics.

### 2.3. Single Photon Emission Computed Tomography

Single photon emission computed tomography (SPECT) is a non-invasive approach based on the property of gastric mucosa to selectively uptake and excrete ^99m^Tc- pertechnetate. Radiolabeled ^99m^Tc is administered intravenously and accumulates in the gastric mucosa, allowing visualization of the entire stomach wall with a SPECT gamma camera [42]. Three-dimensional stomach images are produced by computerized data processing and can be used to calculate stomach volume. The gastric accommodation response after a meal can be estimated by measuring the fundic and total stomach volumes during both fasting and postprandial periods. It has been demonstrated that postprandial volume changes measured by SPECT correspond with barostat volume measurements [43].

SPECT has been well-established and is a validated noninvasive imaging technology used to assess gastric function and structure [31]. However, its wider utilization has been limited due to associated exposure of patients to ionizing radiation, high cost, and relatively limited availability [31]. In addition, SPECT evaluates gastric tone indirectly, which has been shown to be suboptimal in assessing gastric accommodation and cannot simultaneously assess gastric sensation.

### 2.4. Ultrasonography

Transabdominal ultrasonography is another non-invasive method with demonstrated utility and validity for the investigation of gastric accommodation, gastric emptying, and gastroduodenal flow [44]. A conventional ultrasound probe is used to visualize the stomach and can indirectly estimate gastric volumes by measuring the gastric diameter and area [45]. Serial measurements of variations in the area of antral cross-section can serve as an indicator to estimate gastric emptying. Duplex sonography can be used to visualize luminal contents movement [46].

3D ultrasound has also been used to assess gastric structure and function as it provides better visualization of the entire stomach without being limited by anatomical features or air bubbles in the stomach [35]. Compared to SPECT, 3D ultrasound offers an accurate estimation of gastric accommodation [47]. A novel 3D ultrasonography system with automated acquisition was also developed and shown to be reliable for assessing gastric accommodation in healthy adults [48]. Combining ultrasonography with a drinking test (e.g., The Ultrasound Meal Accommodation Test/UMAT) and psychological analysis was reported useful in the evaluation of patients with suspected functional gastrointestinal disease [49]. Advantages of gastric ultrasonography are that it is noninvasive, relatively inexpensive, widely available, and well-tolerated by patients. It is also radiation-free and does not affect gastric motility. Ultrasound limitations include variations in the quality of obtained images dependent on patient’s body habitus or presence of air in the gut, as well as the need for a skilled technician [31]. There is also a lack of standardized values for ultrasound measurements of gastric function and more research is needed to further validate its accuracy and reliability in the assessment of gastric accommodation.

## 3. Assessment of Antroduodenal Motility

### 3.1. Antroduodenal Manometry

Antroduodenal manometry (ADM) allows simultaneous assessment of gastric and duodenal motility by measuring the frequency and characteristics of contraction patterns [50]. In this method, a water-perfused or solid-state (high-resolution) manometric catheter is typically positioned transnasally through the pylorus into duodenum. Consequently, evaluation of the number and amplitude of gastric contractions in both fasting and postprandial states is possible. [51] However, given its invasive nature, ADM is typically applied in selected patients with marked symptoms and presumed severely impaired gastrointestinal function, in whom understanding the pathogenesis of the motility disorder is necessary to guide better treatment [52].

ADM can help identify a major gastric motor disorder, typically found in patients with either neuropathic disorder, myopathic disorder, or chronic intestinal pseudo-obstruction. Postprandial antral hypomotility is another non-specific pattern that can be readily identified by ADM [53]. A reduced postprandial distal antral motility has been associated with delayed solid gastric emptying on a scintigraphic study [54]. In addition to antral motility, duodenal dysmotility, as measured by ADM, was reported to be closely associated with the severity of symptoms in gastroparesis patients [55]. ADM can assist in confirming or excluding an underlying motility disorder in severely symptomatic patients who have no evidence of significantly delayed gastric emptying [56]. Interestingly, a recent study including 167 patients with gastroparesis of different etiologies, from post-surgical, diabetic, to idiopathic, found that the antroduodenal contraction patterns differ among various gastroparesis etiologies [57]. Although suitable for evaluating the effects of pharmacological interventions on gastroduodenal motor functions, there have been few studies in the use of ADM for this purpose [58].

Several pitfalls have limited the broader use of ADM, including difficulties with the catheter placement and patient tolerance of the procedure, as well as the inability to measure the pre- and post-prandial gastric volumes and lack of universally applied and standardized procedural protocols [52]. For these reasons, ADM is only available in few selected referral tertiary care centers where adequate expertise can be offered.

### 3.2. Magnetic Resonance Imaging

Dynamic magnetic resonance imaging (MRI) is a noninvasive technique that uses body-surface mapping and assessment of contractions to simultaneously analyze gastric motility, anatomy, and emptying with high spatial and temporal resolution [59]. Gastric MRI requires consumption of a test meal containing paramagnetic ions, such as manganese and gadolinium, to allow adequate reconstruction of stomach images [31]. Using both liquid and solid meals, gastric MRI has been validated to accurately investigate the gastric volume content and emptying rate [60]. An MRI acquisition protocol and image processing pipeline were proposed in a recent study to assess gastric wall motion and emptying in healthy adults. [61] This methodology enables direct visualization of peristaltic waves along the gastric wall and can provide personalized profiles of gastric contractions and emptying.

The advantages of MRI as a method to assess gastric motility are its high-resolution, no patient exposure to radiation, good soft-tissue contrast, and the ability to assess gastric contractions and emptying simultaneously [52]. The primary challenges of MRI methodology include high cost and time required to complete the scan and analysis, as well as the impact of body position and respiratory motion on the quality of obtained images. Furthermore, clinically applicable referent values and standardized analysis software need to be developed and validated.

## 4. Assessment of Gastric Emptying

### 4.1. Gastric Emptying Scintigraphy

Gastric emptying scintigraphy is a conventional method to measure the rate of gastric emptying. It is considered the gold standard. A standardized meal of liquid egg whites (approximately 240 kcal, 2% fat) is mixed with ^99m^Tc sulfur colloid and cooked until it has a texture of firm omelet. [62]. After ingestion of the meal, the gastric area is scanned with an antero-posterior γ-camera over 4 h. (unless over 90% of the solid meal has already emptied at 3 h) [62]. The normal values of gastric emptying have been established and internationally validated with gastric solid meal retention of >10% at 4 h post-ingestion; this is considered a delayed gastric emptying [63]. This methodology was proven reproducible when applied in patients with upper gastrointestinal symptoms [64]. Although the relationship between gastric emptying rate and gastrointestinal symptoms has been controversial [65], studies using scintigraphy with a solid meal and collecting data for at least 3 h, post-ingestion, showed positive correlation between gastric emptying rate and severity of nausea, vomiting, abdominal pain, and early satiety/fullness [65]. Consequently, the current American College of Gastroenterology (ACG) clinical guideline for gastroparesis recommends gastric emptying scintigraphy, measuring solid meal emptying over a period of at least 3 h, as a first-line test to establish the diagnosis of gastroparesis in patients with clinical presentations suggestive of gastroparesis [3].

In addition to assessing the rate of gastric emptying, attempts were made to implement gastric emptying scintigraphy as a tool to evaluate gastric accommodation. A recent study proposed the use of scintigraphy-based radiolabeled meal intragastric distribution immediately after ingestion for this purpose [66]. However, in patients with diabetes and upper gastrointestinal symptoms, the intragastric meal distribution did not significantly correlate with gastric accommodation measured by SPECT. Additional validation is necessary before the scintigraphic measurement based on intragastric meal distribution can be used for assessment of gastric accommodation [67].

Advantages of gastric emptying scintigraphy as a clinical tool to measure gastric emptying are that it is a relatively available, well-validated, non-invasive test with the ability to depict post-meal food distribution in the stomach. However, gastric emptying scintigraphy also has limitations, including: (1) radiation exposure limiting its applicability (e.g., women of childbearing potential, reduced ability to repeat studies, etc.); (2) the standardized meal with its low caloric and fat content may not correspond to a typical meal consumed by symptomatic patients and therefore not induce equivalent gastric physiology, resulting in gastric emptying scintigraphy potentially underdiagnosing gastroparesis; (3) relatively long time required to adequately complete the test limits the availability of needed equipment (e.g., γ-camera) and the number of patients that can be investigated. Moreover, it has been reported that, when repeated, gastric emptying scintigraphy results might be relatively labile. In a study by the Gastroparesis Clinical Research Consortium, 42% of patients initially diagnosed with gastroparesis, and 37% of the patients first diagnosed with FD (no delayed gastric emptying), were reclassified after completing a repeat gastric emptying scintigraphy [68]. This study concluded that the diagnosis of FD and gastroparesis is interchangeable and that the pathophysiology of symptoms is not consistently captured by gastric emptying findings, which has long been a gold standard in the field. Based on these findings, authors proposed that both FD and gastroparesis should be considered part of the same spectrum of truly “organic” gastric neuromuscular disorders, prompting us to reevaluate the importance of gastric emptying test in symptomatic patients.

### 4.2. Gastric Emptying Breath Test

The stable isotope breath test methodology uses a standardized solid meal containing *Spirulina platensis* (edible blue-green microalgae) tagged with a non-radioactive carbon isotope ^13^C^50^. After ingestion, metabolites of spirulina are absorbed in the small intestine and metabolized by the liver, resulting in ^13^C excretion in the lungs. Isotope ratio mass spectrometry is used to determine the rise of ^13^C over baseline in exhaled breath samples. This rise correlates with the rate of gastric emptying and allows indirect calculation of the gastric emptying time [69]. Studies of the liquid- and solid-phase gastric emptying have used labeled acetate and octanoate, respectively [70]. ^13^C-glycine was also selected as a marker for the liquid phase because it is rapidly absorbed and transformed into ^13^CO_2_ after reaching the small intestine [71]. The current ACG clinical guideline considers gastric emptying breath test a reliable method to assess gastric emptying in patients with suspected gastroparesis [3]. The ^13^C spirulina breath test has also been approved by the US Food and Drug Administration (FDA) [52]. However, in practical terms, this test is still not widely available to many providers in the US. The ^13^C breath test performed simultaneously with scintigraphy was shown to be reproducible and correlate well with gastric emptying scintigraphy in both healthy volunteers and symptomatic patients with suspected gastric emptying delay [69].

The advantages of gastric emptying breath testing are that it is non-invasive and has no radiation exposure. This technique can be performed at the point of care since obtained breath sampless are stable and can be stored for a prolonged period, allowing centralized analysis at a later time. The primary limitation of the gastric emptying breath test is that it is an indirect assessment, and that its accuracy can be affected by liver, lung, or malabsorptive diseases. This is relevant because malabsorptive syndromes such as lactose malabsorption or small intestinal bacterial overgrowth were reported in common with gastroparesis [72], particularly in patients with predisposing conditions such as diabetes or scleroderma [73].

### 4.3. Wireless Motility Capsule

The wireless motility capsule (WMC) has been developed as a non-invasive test and alternative to scintigraphy, used to measure gastric emptying as well as small and large intestine transit time [52]. The test (SmartPill GI Monitoring System) has been approved to assess gastric emptying by the US Food and Drug Administration. It is an indirect test where a swallowed capsule records and transmits the luminal pH, pressure, and temperature to an external recorder. The gastric emptying time is derived by a rise of pH from the acidic low gastric baseline to values above 4 in the duodenum [74]. It has been shown that the capsule, when ingested with a meal, is emptied from the stomach separately from the meal through reactivation of the interdigestive migrating motor complex [75]. The gastric emptying time observed by the WMC was found to significantly correlate (r = 0.73) with the gastric emptying at 4 h derived by scintigraphy in both healthy and gastroparetic individuals [74]. The WMC gastric emptying cut-off time of 300-min had a sensitivity of 65% and a specificity of 87% in comparison with scintigraphy at 4 h [74]. In a recent study, WMC reported delayed gastric emptying in twice as many patients with diabetes when compared to the evaluation with gastric emptying scintigraphy [76], but the clinical relevance of higher sensitivity for WMC to detect delayed gastric emptying is not clear. WMC distinguished gastroparesis from healthy subjects based on the motility profiles of the stomach and small bowel [77]. However, no correlation was observed between detection of gastric emptying delay on WMC and the overall gastroparesis symptoms [78].

A major advantage of WMC is the ability to evaluate motility and transit times for the whole gut, including separate assessments of the stomach and small and large intestine [79]. As up to 65% of patients with gastroparesis may have slow transit constipation [80], the WMC may provide additional value in the clinical management of patients with both gastroparesis and constipation symptoms. Other advantages of WMC are that it is well tolerated by patients, there is no associated radiation exposure, and its overall clinical safety. However, the WMC should not be performed if mechanical obstruction is suspected and caution should be undertaken if considered for patients with post-surgical gut anatomy [53]. It also has a relatively high cost which limits its availability in routine clinical practice.

## 5. Assessment of Gastric Myoelectrical Activity

### 5.1. Electrogastrography

Transcutaneous electrogastrography is a non-invasive technique utilizing skin electrodes placed on the abdomen to capture gastric myoelectrical activity. Typically, the test is performed in a quiet room while the studied individual is in a supine position. To achieve reliable results, the gastric myoelectrical activity should be recorded for at least 15 min in fasting state and 30 min in postprandial state. These recordings generate the electrogastrogram (EGG). The test meal should contain at least 250 kcal (preferably > 400 kcal), with a maximum fat content of 35% [81]. The EGG has been validated to detect the slow electrical gastric rhythm, termed slow waves [82], that determine the propagation and maximum frequency of gastric contractions.

Major EGG parameters include the dominant frequency, dominant power, and percentage of normal gastric slow waves as well as percentage of gastric electrical dysrhythmia, including tachygastria, bradygastria, and arrhythmia [83]. The frequency of gastric electrical slow waves in healthy individuals is ~3 cycles per min (cpm) with a normal range of 2 to 4 cpm. Bradygastria is defined as slow waves with a frequency of <2 cpm, and tachygastria is defined as slow waves with a frequency of >4 cpm. Previous studies have shown that the dominant power of the EGG is correlated with antral contractions [84,85,86] and abnormal slow waves assessed from the EGG are associated with delayed gastric emptying [87]. Accordingly, the EGG can be used as a noninvasive alternative in assessing gastric motility.

Advantages of electrogastrography are that it is relatively simple to complete, non-invasive, and its technical quality has significantly improved over the last 20 years. While there are concerns about the accuracy of the EGG, there is no question that the EGG does reflect gastric myoelectrical activity. However, caution should be exercised during data acquisition and interpretation. Motion artifacts should be minimized during the recording and possible noise or interference in the EGG should be taken into consideration while interpreting the EGG data.

### 5.2. High-Resolution Electrogastrography

High-resolution electrograstrography (HR-EGG) has emerged as a novel, non-invasive methodology offering improved capture of the gastric myoelectrical activity, such as propagation of slow waves and spatial abnormalities of gastric slow waves. In HR-EGG, an array of densely spaced cutaneous electrodes is placed on the abdomen over the stomach [88] to record multichannel EGG signals with a high spatial resolution. In one recent study, the HR-EGG revealed spatial slow wave abnormalities in 44% of patients with upper GI symptoms and there was correlation of aberrant slow wave propagation direction with gastroparesis symptoms and abdominal pain [89]. However, more studies are needed to overcome technical challenges (e.g., sufficiently high-density grid of electrodes to account for gastric anatomical variability while allowing appropriate skin contact) and to establish the clinical role of HR-EGG in diagnosing gastric motility disorders and assessing the electrophysiology of the stomach.

### 5.3. Body Surface Gastric Mapping

Body surface gastric mapping (BSGM) represents a further refinement of the technology used to optimally capture the gastric myoelectrical activity in non-invasive manner. In BSGM, a comprehensive spatial analysis of gastric potentials, obtained via dense grid of cutaneous electrodes, is generated to derive detailed maps of gastric patterns. Several technical advancements, such as sophisticated bioamplifiers and signal processing systems specifically aimed for gastric electrophysiology with validated ability for artifact detection and rejection, make BSGM a significant improvement over the earlier EGG modalities [90,91]. This methodology may enhance the current clinical utility of EGG by allowing more comprehensive gastric slow wave analysis and better characterization of gastric dysrhythmias [92]. For example, a recent study by Gharibans et al., using a BSGM device to explore the pathophysiology of chronic nausea and vomiting, found that BSGM results are capable of identifying specific disease phenotypes in these patients [23]. Furthermore, the US FDA recently approved the first commercially available device for performing BSGM. This device should enable a broader use of this technology. However, to provide consistent reporting, BSGM reference ranges are still being established and, although normative values were recently proposed [93], more studies are needed to establish the clinical role of BSGM when evaluating gastric motility disorders and when assessing the electrophysiology of the stomach.

## 6. Discussion

Adequate and accurate evaluation of gastric motility is critically important for diagnosis and management of patients with functional dyspepsia and gastroparesis, disorders that affect a large proportion of the general population. Impaired gastric accommodation, antral hypomotility, delayed gastric emptying and gastric electrical dysrhythmia are major pathophysiological mechanisms in functional dyspepsia and gastroparesis [94]. In this mini-review, we summarized various methods currently used for assessment of gastric motility including their advantages and limitations. The diagnostic methods illustrated in this review are shown in Figure 2.

Various methods are available for assessment of gastric accommodation. While the gastric barostat method is considered a gold standard, it is not applied in clinical practice and used infrequently in clinical research due to its invasiveness and poor tolerance by patients. However, the barostat remains a valuable tool for assessing gastric accommodation and sensitivity in animal models [95] and experimental human studies focusing on gastric physiology. Each of the alternative methods for assessing gastric accommodation has unique advantages and limitations. The satiation drinking test has emerged as a non-invasive, clinically applicable, and reproducible test. This test was proposed a diagnostic biomarker for FD, given its high correlation with hallmark symptoms of FD, and a possible biomarker for therapeutic response in FD [96]. However, widespread clinical use of the satiation drinking test will require further support from clinical research, validating its ability to predict therapeutic responses to specific interventions in FD. SPECT is a non-invasive and validated technology used to assess the gastric accommodation and postprandial gastric volume changes. Still, its role is likely to remain primarily in research, given associated ionizing radiation, high cost, and limited availability, restricting broader clinical application. Abdominal ultrasonography is an appealing method to evaluate the gastric accommodation in clinical settings given its non-invasiveness, safety, and broad availability. It can be used in conjunction with the satiation drinking test [97] to assess the drinking capacity of FD patients. Clinical studies are needed to validate this approach and evaluate the ability of the abdominal ultrasonography combined with satiation drinking test to develop into a therapeutic biomarker in patients with upper gastrointestinal symptoms.

Antroduodenal manometry is a traditional method for assessment of antral and duodenal contractions. Due to its invasiveness and requirement for prolonged recording, it is currently performed in only a relatively small number of tertiary medical centers. ADM was found useful in an animal model of gastroparesis [98]. It has a clinical role in patients with intestinal pseudo-obstruction, severe unexplained nausea and retching, and systemic sclerosis. ADM has also been used in clinical practice to assist selection of dietary recommendations and predict ability to tolerate enteral feeding in severely symptomatic patients with suspected motility disorder [52,99]. Expert consensus considers ADM a reference technique for the analysis of gastric contraction patterns [52]. However, more case-controlled investigations are needed to solidify the relevance of antroduodenal motility patterns captured using manometry in symptoms induction and management. If available and/or affordable, gastric MRI is an attractive method for assessment of gastric contractions. The major advantage of gastric MRI is that it can be used to simultaneously assess gastric accommodation, gastric emptying, and antral contractions [100]. Gastric MRI has been used in basic research studies to assess gastric physiology in rats [101]. Studies have begun to explore the clinical role of MRI in patients with functional gastrointestinal disorders, pseudo-obstruction, and inflammatory bowel disease [102]. Currently, however, there is a lack of clinically validated normative values for various MRI data components to assess the gastric motility. More research is needed for comprehensive gastric MRI data analysis, including further refinement of the software and standardization of MRI scanning protocols. Clinical studies are also needed to investigate its role in assessing gastric motility functions in patients with functional dyspepsia and gastroparesis.

Gastric emptying scintigraphy is the most commonly used method for assessment of gastric motility in clinical practice. Scintigraphy is readily available and noninvasive. Testing protocols and normative values are well-established for the diagnosis of delayed gastric emptying. Scintigraphic gastric emptying assessment is the test of choice, recommended by societal clinical guidelines, to diagnose gastroparesis [103]. One limitation is the lack of correlation between scintigraphic gastric emptying and major symptoms of gastroparesis, although this may be related to the adequacy of applied scintigraphic protocol [96]. Dedicated animal pinhole scintigraphy has also been used for measurement of gastric emptying in small laboratory animals such as mice. Scintigraphic gastric imaging may be further enhanced by more subtle data analysis, such as meal distribution in different segments of the stomach, and the addition of complementary methods to better determine the pathophysiological factors associated with symptoms. Gastric emptying breath test is relatively easy to perform and radiation-free. The stable isotope breath test was shown accurate for evaluation of patients with suspected gastroparesis [103]. However, although relatively available, it is not yet widely adopted in clinical practice. ^13^C-Acetic acid breath test was also proven useful in animal studies for evaluation of gastric emptying changes [104]. WMC provides an exciting opportunity to evaluate the motility of the entire gastrointestinal tract during a single study. WMC is considered a viable alternative to gastric emptying scintigraphy for evaluation of gastroparesis by the ACG guideline [3]. WMC has also been used in animal studies for comprehensive assessment of the canine gastrointestinal transit [105]. The clinical application of WMC may be particularly attractive for patients with concomitant upper and lower gastrointestinal syndromes as it allows comprehensive assessment of the gastrointestinal motility. Additional studies are needed to further delineate the relationship between WMC measurements and symptom syndromes suggestive of multiregional or generalized gastrointestinal dysmotility.

The EGG/HR-EGG, when recorded appropriately, is a reliable methodology used to delineate gastric electrical slow waves. Since the gastric slow wave is the basic electrical rhythm that determines the propagation and frequency of gastric contractions, detection of gastric electrical dysrhythmia is physiologically important and of clinical significance. The major disadvantage of EGG is that it is sensitive to motion artifacts [106]. Caution and expertise should be used when recording and interpreting the EGG. High-resolution EGG is still being studied, and this research should assist in defining its clinical role, especially in patients with FD and gastroparesis. EGG recordings allow detection of the effect of prokinetic medications on gastric myoelectrical activity [107]. BSGM is the most recently developed method of capturing the gastric myoelectrical activity with technical improvements that allow more optimal collection of gastric signals with minimal artifacts contamination. Further studies to establish the normative values for BSGM data and to better characterize its clinical role are needed.

Another important aspect when considering various methods to evaluate gastric motility is the relative cost of individual techniques. While this is highly variable and may depend on variety of factors, in general, gastric motility tests can be stratified into the following three categories: (1) relatively inexpensive (e.g., satiation drinking test, EGG), (2) moderately expensive (e.g., ultrasonography, barostat testing, gastric emptying breath testing), and (3) expensive methods (e.g., gastric emptying scintigraphy, wireless motility capsule, antroduodenal manometry and MRI or SPECT imaging).

There are additional emerging diagnostic methods for the assessment of gastric motility that currently are in the investigational stage and not yet clinically applicable. For example, functional lumen imaging probe (EndoFLIP) is a developing method that can be applied during upper gastrointestinal endoscopy to assess the pyloric diameter and distensibility [108]. Limited data suggest that functional lumen imaging of the pylorus may help identify those gastroparesis patients who will benefit from pylorus-directed interventions such as gastric peroral endoscopic myotomy (G-POEM) or intra-pyloric botulinum toxic injection [109]. Brain functional MRI (fMRI) imaging has also been recently used to link altered brain activity and connectivity with examinations of gastric physiologic functions [110]. Each of these methods may allow enhanced understanding of gastric motility and its contribution to the pathogenesis and treatment of disorders such as functional dyspepsia and gastroparesis. In summary, gastric motility assessment is critically important to better understand the pathophysiology and guide the therapy of common disorders of gut-brain interaction such as functional dyspepsia and gastroparesis. Although various assessment methods are currently available, improved understanding of the clinical yield from obtained data in relation to patient outcomes is needed. Further development of more advanced and less invasive methods may also allow wider applicability and clinical use.

## Figures and Tables

**Figure 1 diagnostics-13-00803-f001:**
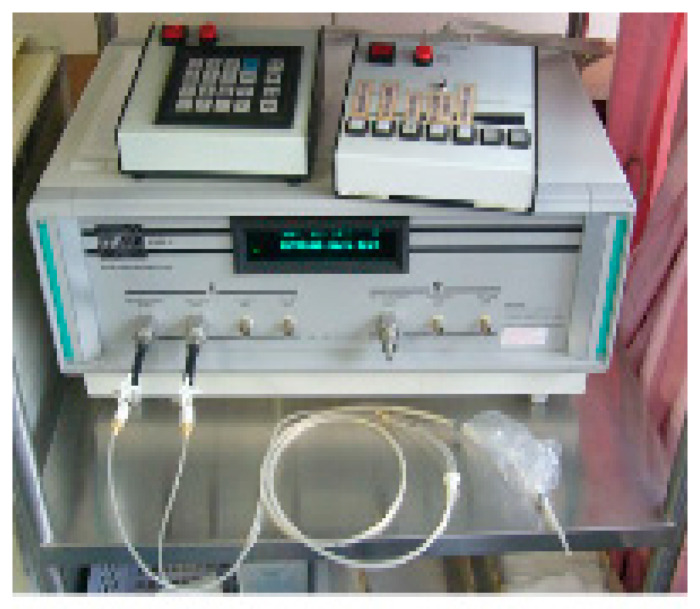
A barostat device used to assess gastric accommodation.

**Figure 2 diagnostics-13-00803-f002:**
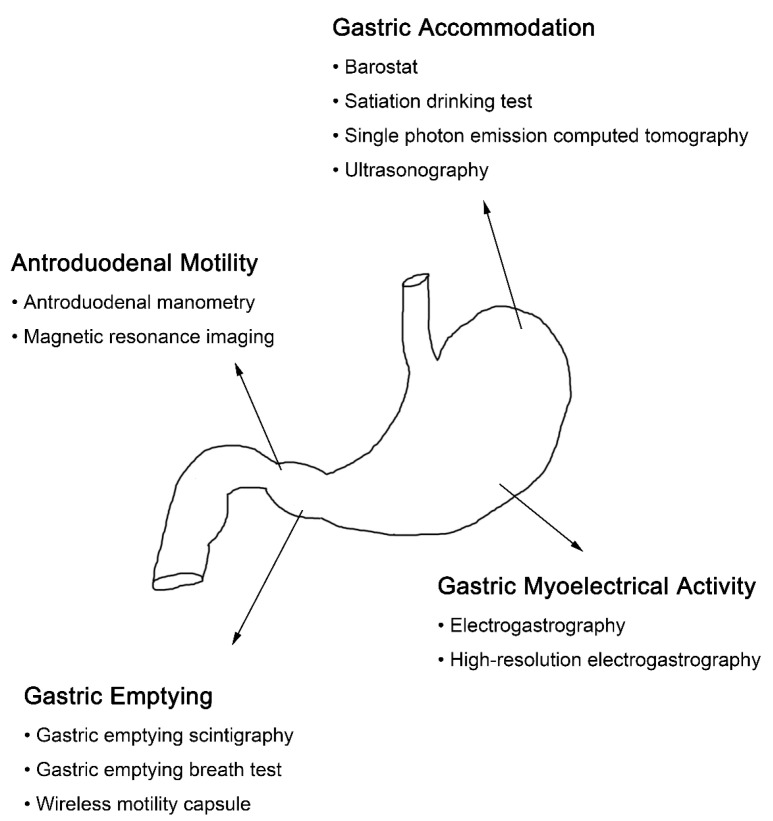
Summarized diagnostic methods for evaluation of gastric motility.

## Data Availability

Not applicable.

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
