# Peer review of "Diagnostic Methods for Evaluation of Gastric Motility—A Mini Review"

_diagnostics, 2023, doi:10.3390/diagnostics13040803_

Round 1
Reviewer 1 Report
In this paper, the authors provide a review of the various diagnostic methods for evaluating gastric motility that are currently in use in clinic. In doing so, they provide a summary of each technique, as well as various advantages and disadvantages.
The paper is very well written, and was easy to read and follow. I enjoyed reading this paper - it is a very good review and will be valuable to the field, particularly to students and trainee clinicians. I found the review to be relatively comprehensive, particularly for a "mini-review" as the title suggests.
I have included a few suggestions that the authors may wish to include.
General comments:
1) The addition of a section on emerging diagnostic methods that are still in the research phase and have not yet matured to clinical uptake would help to expand the breadth of the review and provide readers with a snapshot of some of the exciting new methods and techniques being explored. I don't think it needs to be a long section, but a brief description of emerging techniques with references would likely be of interest to the readers.
2) How do HR-EGG and body surface gastric mapping differ? What are the strengths and limitations of body surface gastric mapping, and where could it lead in the future? E.g., Carson et al, 2022, NGM, doi: 10.1111/nmo.14048; Gharibans et al 2022, Sci Trans Med, doi: 10.1126/scitranslmed.abq3544. The addition of a section of body surface gastric mapping may be appropriate and could fit well after the section on high-resolution EGG?
3) How does the relative cost of each of the various techniques compare? The only place where the cost was mentioned was for the wireless motility capsule. It would be valuable to comment on the relative cost of the other methods.
4) There are no references in the entire Discussion section, which is quite unusual. It would be good to see some of these statements supported with relevant references, particularly for readers wanting more information on some of the specific techniques and/or discussion here. It may also be valuable to refer the readers to other good review papers here that provide a comprehensive review on one or more of these techniques.
Specific comments:
5) There are a lot of gaps in the text where there are extra spaces. This occurs multiple times on almost every page of the manuscript, and I am unsure of the cause.
6) Page 1, last paragraph: It may be helpful to specify "gastric electrical dysrhythmia" instead of just "gastric dysrhythmia".
7) Page 2, paragraph 2: it may be helpful to similarly specify "gastric electrical slow waves", instead of just "gastric slow waves".
8) Page 2, paragraph 4: The authors may wish to consider including mention of the very recent paper by Gharibans et al that demonstrated dysrhythmias associated with nausea and vomiting syndromes using HR-EGG / body surface gastric mapping (2022, Sci Trans Med, doi: 10.1126/scitranslmed.abq3544).
9) Page 2, paragraph 4: The authors may wish to include studies here by the O'Grady group that have correlated ICC depletion with abnormal gastric myoelectrical activity (dois: 10.1053/j.gastro.2012.05.036; 10.1053/j.gastro.2015.04.003).
10) Page 2, paragraph 5: "L" is the standard abbreviation for liters, not "Lt". It would be good to check abbreviations and units throughout the manuscript. For example, I noted that "hours" was typically used, but "h" was occasionally used instead.
11) Page 3, paragraph 1: temporary gastric electrical dysrhythmias were recently found to be induced by gastric distension with a barostat device, which may be relevant here. Doi: 10.1152/ajpgi.00219.2021.
12) Pages 5-6: I find the abbreviation of gastric emptying scintigraphy as "GES" to be confusing because GES is also typically used to abbreviate "gastric electrical stimulation". It seems that GES is indeed a commonly used abbreviation for both of these techniques, but it may be confusing to other readers as well.
13) Page 6, paragraph 1: "clinical presentations suggestive gastroparesis" >> "clinical presentations suggestive of gastroparesis".
14) Page 6, paragraph 3: I was surprised to see reference 62 (Pasricha et al 2021, Gastroenterology, doi: 10.1053/j.gastro.2021.01.230) to be referred to as a minor comment at the end of this section. This is a major recent study by the GPCRC that highlights the relative unreliability of gastric emptying tests, which have long been a gold standard in the field, including by the GPCRC. It seems that this recent study and shift away from gastric emptying tests should be given more prominence in this section?
15) Page 7, paragraph 4: the text currently states that "Bradygastria is defined as slow waves with a frequency of <3 cpm", but this should instead be "<2 cpm".
16) The 'Author Contributions' section has not been completed and is still the generic template text. The following 3 sections have also not been completed and are still the generic template text (in red), but are presumably not applicable to this review paper and can therefore be removed (IRB Statement, Informed Consent Statement, and Data Availability Statement).
17) Ref 73 has errors - author is missing, title is all caps, and journal title is not abbreviated.
Reviewer 2 Report
In this mini-review the Authors discuss diagnostic methods for the evaluation of gastric motility. The paper is well organized, but some points should be raised.
Information on functional dyspepsia and gastroparesis are preliminarly provided but the main difference to guide the correct diagnosis between these two conditions is not indicated.
Please add this point in the “Introduction” section.
Gastric barostat is considered the gold standard for gastric motility evaluation. However, in this mini-review, the text dealing with this diagnostic test is the shorter one! Very few information on its clinical and research applications are provided.
Please, add information on glycine, octanoate and acetate 13C breath test for gastric emptying rate.
Please, underline the questionnable accuracy of EGG: we are still not sure that recordings obtained with this instrument are related to gastric myoelectrical activity.
The “Discussion” section is just a summary of what reported in the review. The Authors should indicate how these text can be used in clinical practice, which test is included in clinical guidelines, which test should be considered suitable for basic evaluation and which one for a more accurate or research-based studies. The reader should be guided through the review in the learning of the available texts but the Authors should also indicate their clinical application.
Round 2
Reviewer 2 Report
The paper was much improved in the revised form. I have no further comments, even if I still have some concerns on the accuracy of EGG: I strongly disagree on the absence of doubts on what EEG measures...